



**Deriving a country-wide soils dataset from the Soil Landscapes of Canada (SLC) database**
**for use in Soil and Water Assessment Tool (SWAT) Simulations**
Marcos R. C. Cordeiro[1], Glenn Lelyk[2], Roland Kröbel[1], Getahun Legesse[3], Monireh Faramarzi[4],
M. Badrul Masud[4], Tim McAllister[1]
[1] Science and Technology Branch, Agriculture and Agri-Food Canada, Lethbridge, AB, T1J 4B1,
Canada
[2] Science and Technology Branch, Agriculture and Agri-Food Canada, Winnipeg, MB, R3T
2N2, Canada
[3] Department of Animal Science, University of Manitoba, Winnipeg, MB, R3T 2N2, Canada
[4] Department of Earth and Atmospheric Sciences, University of Alberta, Edmonton, AB, T6G
2E3, Canada
*Correspondence to*: Tim McAllister (Tim.McAllister@agr.gc.ca).
**Abstract**
The Soil and Water Assessment Tool (SWAT) model has been commonly used in Canada for
hydrological and water quality simulations. However, pre-processing of critical data such as soils
information can be laborious and time-consuming. The objective of this work was to pre-process
the Soil Landscapes of Canada (SLC) database to offer a country-level soils dataset in a format
ready to be used in SWAT simulations. A two-level screening process was used to identify
critical information required by SWAT and to remove records with information that could not be
calculated or estimated. Out of the 14,063 unique soils in the SLC, 11,838 soils with complete
information were included in the dataset presented here. Important variables for SWAT
simulations that are not reported in the SLC database [e.g. hydrologic soils groups (HSGs) and
erodibility factor (K)] were calculated from information contained within the SLC database.
These calculations, in fact, represent a major contribution to enabling the present dataset to be



used for hydrological simulations in Canada using SWAT and other comparable models.
Analysis of those variables indicated that 21.3 %, 24.6 %, 39.0 %, and 15.1 % of the soils in
Canada belong to HSGs 1, 2, 3, and 4, respectively. This suggests that almost two-thirds of the
soils have a high (i.e., HSG 4) or relatively high (i.e., HSG 3) runoff generation potential. A
spatial analysis indicated that  20.0, 26.8, 36.7 and 16.5 % of soil belonged to  HSG 1, HSG 2,
HSG 3, and HSG 4, respectively. Erosion potential, which is inherently linked to the erodibility
factor (K), was associated with runoff potential in important agricultural areas such as southern
Ontario and Nova Scotia. However, contrary to initial expectations, low or moderate erosion
potential was found in areas with high runoff potential, such as regions in southern Manitoba
(e.g. Red River Valley) and British Columbia (e.g. Peace River watershed). This dataset will be a
unique resource to a variety of research communities including hydrological, agricultural and
water quality modellers and are publicly available at doi:10.1594/PANGAEA.877298.
KEY WORDS: Modelling, SWAT, input datasets, soils, Canada.

**1.  Introduction**

Integrated environmental modeling is inspired by modern environmental problems and
enabled by transdisciplinary science and computer capabilities that allow the environment to be
considered in a holistic way (Laniak et al., 2013). In an agricultural context, synthesis and
quantification of multi-disciplinary knowledge via process-based modeling are essential to
manage systems that can be adapted to continual change (Ahuja et al., 2007). The Soil and Water
Assessment Tool (SWAT) (Arnold et al., 1998) is an example of such a process-based model. It
has been developed over the past 30-years to evaluate the effects of alternative management
decisions on water resources and nonpoint-source pollution in large river basins through the
simulation of major processes including hydrology, soil temperature and properties, plant



growth, nutrient and pesticides dynamics, bacteria and pathogens transport, and land
management (Arnold et al., 2012; Douglas-Mankin et al., 2010). Furthermore, a weather
generator is included in the model to fill gaps that may exist in meteorological records.
The SWAT model has been extensively tested around the world for a wide range of hydro-
climatic conditions, water and land management practices, and time scales (Douglas-Mankin et
al., 2010). The wide adoption of the SWAT model has been prompted by pre- and post-
processing software tools such as a GIS interface, extensive user documentation (Arnold et al.,
2012), as well as several linked databases for crops, soils, fertilizers, tillage, and pesticides
(Santhi et al., 2005). Among these, soil properties are especially important as they are needed for
the simulation of influential processes such as evapotranspiration, soil water balance, nutrient
dynamics, and sediment transport (Neitsch et al., 2005). However, the existing built-in database
is only valid for SWAT applications in the USA. Accordingly, studies outside the USA require
the development of a soils dataset by pre-processing available soils data into a format readable
by SWAT, a time consuming process as not all data required by SWAT is readily available for
countries outside of the USA.
In Canada, the SWAT model has been used for hydrological simulations in most provinces,
including Prince Edward Island (Edwards et al., 2000), New Brunswick (Chambers et al., 2011;
Yang et al., 2009), Nova Scotia (Ahmad et al., 2011), Ontario (Asadzadeh et al., 2015; Rahman
et al., 2012), Quebec (Lévsque et al., 2008), Manitoba (Yang et al., 2014), Saskatchewan
(Mekonnen et al., 2016), Alberta (Mapfumo et al., 2004; Watson and Putz, 2014; Faramarzi et
al., 2015), and British Columbia (Zhu et al., 2012). However, preparation of Canadian soils
information in a consistent and usable format for SWAT is time consuming (Rahman et al.,
2012), as information has to be collected from soil reports, cross-checked against GIS datasets,



missing soil variables have to be calculated from other physical and hydraulic properties, and all
parameters have to be attributed to specific soil grids or polygons.

Some of this pre-processing work can be alleviated by using publically available databases

that contain most of the information required by SWAT. The Soil Landscapes of Canada (SLC)
database published by Agriculture and Agri-Food Canada (Soil Landscapes of Canada Working
Group, 2010) is an example, and has been used in SWAT applications in Ontario (Asadzadeh et
al., 2015; Rahman et al., 2012), Saskatchewan (Mekonnen et al., 2016), Alberta (Faramarzi et al.,
2015), and British Columbia (Zhu et al., 2012). The SLC contains a series of GIS dataset that
provides information about the country's agricultural soils at the provincial and national levels. It
was compiled at a scale of 1:1 million, and the information is organized according to a uniform
national set of soil and landscape criteria based on permanent natural attributes (Soil Landscapes
of Canada Working Group, 2010). The SLC encompasses the southern portions of the Provinces
of Ontario and Quebec and a larger portion of the Prairies Provinces of Manitoba, Saskatchewan,
and Alberta as far north as to the boreal shield. Coverage in the maritime provinces of New
Brunswick, Nova Scotia, and Prince Edward Island is nearly complete (Fig. 1).

Although there are more detailed soil datasets available at provincial levels (e.g. AGRASID

dataset in Alberta), selection of SLC for integration with SWAT was based on the fact that i) it
covers all of Canada's agricultural soils in a single dataset; ii) it has been used in regional studies
in Canada, as described above; and iii) it is more easily applicable to large-scale national studies
as broad-scale datasets require reduced resources to prepare and process data (Moriasi and
Starks, 2010). Modelling studies comparing the performance of a single model (calibrated and
un-calibrated) but using soil datasets with varying spatial resolution in the USA [i.e., the State
Soil Geographic database (STATSGO) compiled at 1:250,000 scale, and the Soil Survey





Geographic database (SSURGO) with scales ranging from 1:12,000 to 1:63,360] also revealed
that using either dataset produced comparable results (Mednick, 2008).

Due to the importance of the SWAT model for integrated environmental modeling in

Canada, and the prominence of the SLC database as a potential input dataset for this model at a
national level, the objective of this work was to offer a country-level soils dataset in a format
ready to be used in SWAT simulations. The dataset was derived to provide over 20 parameter
values for different soil types that are varied for each soil layer. It was prepared in a format
suitable for use in the ArcSWAT version of the model, which is attributed to a grid or polygon-
based soil map. Such a laborious pre-processing exercise is widely, but inconsistently adopted in
SWAT simulations reported in the literature. Finally, deficiencies in the dataset are also
presented and discussed.
**2.  SLC data structure**
The SLC database (http://sis.agr.gc.ca/cansis/nsdb/slc/v3.2/index.html) is structured as a
component-based GIS layer, where a single polygon may contain several soil series. This
structure is similar to that of the State Soil Geographic (STATSGO) database in the United
States (Srinivasan et al., 2010). Such structure creates a one-to-many relationship where the
multiple soil components of a polygon are not spatially defined. The actual soil information in
the SLC database is stored in a number of tables linked together through intricate relationships
(Soil Landscapes of Canada Working Group, 2010). Among these, four tables are relevant for
developing a dataset for SWAT applications:
• the Polygon Attribute Table (PAT) provides the linkage between geographic locations

(polygons in the SLC GIS coverage) and soil landscape attributes in the associated



database tables (e.g. unique soil ID in the SNT and respective number of layers in the
SLT);
• the Component Table (CMP) describes each of the individual soil and landscape

features comprising the polygons. That is, it describes which soil(s) are present in

each spatial unit (i.e., polygon) in the GIS layer;

•  the Soil Name Table (SNT) describes the general physical and chemical

characteristics for all of the soils identified in a geographic region;

• the Soil Layer Table (SLT) contains soil information that varies in the vertical

direction (i.e., layered attributes).

The CMP table describes the proportion of each non-spatially defined soil component in a
polygon if more than a soil component exists [the soil component(s) refer to the soil(s)
element(s) that comprise each polygon]. The component numbering follows a sequence of
decreasing proportion in a polygon (i.e., first component has the highest proportion; last
component has the smallest proportion). This component-based structure of the SLC database
does not affect the analysis since all the soils listed in the SNT table were processed to generate
the present dataset. However, it has implications for the SWAT model user, who has to make a
decision on how to handle the relationship between the polygon (spatially defined) and each non-
spatially defined soil component in multi-component polygons (e.g. selecting the larger
component in a polygon or generating a hybrid soil incorporating properties of each soil
component).
**3. SWAT soils data structure**
The SWAT soils information is stored in the 'usersoil' table, located within the SWAT 2012
database in Microsoft Access format (i.e., SWAT2012.mdb). Each soil is stored as a new record



(i.e., row) in the table. Specific soil variables (Table 1) comprise the 152 columns of the user soil
table. The first column is an OBJECTID field assigning an unique identifier for each record.
Columns two through six pertain to soil classification. The second column is the map unit
identifier (MUID), which is used for mapping a collection of areas grouped by the same soil
characteristics. A single MUID may describe different soil types, which are stored with a record
counter in the third column (SEQN), while a soil identifying name (SNAM), a soil interpretation
record (S5ID), and the percent of each soil component (CMPPCT) are recorded in the fourth,
fifth, and sixth columns, respectively (Sheshukov et al., 2009). Columns seven through twelve
describe major soil properties pertaining to the soil type, namely, the number of layers
(NLAYERS), the hydrological soil group to which that soil belongs (HYDGRP), the maximum
rooting depth of the soil profile (SOL_ZMX), the fraction of soil porosity from which anions are
excluded (ANION_EXCL), the potential of maximum crack volume of the soil profile expressed
as a fraction of the total soil volume (SOL_CRK), and the texture of the soil layer (TEXTURE).

The next 120 columns starting from column 13 (i.e., columns 13 to 132) describe the

information for each layer of the soil profile. These columns are arranged in sets of 12 variables
each for 10 possible soil layers. The variable NLAYERS indicates how many of these sets
should be populated. Variables for any sets beyond NLAYERS should be assigned a value of
zero. The variables included in each set of soil layers are the depth from soil surface to bottom of
layer (SOL_Z), moist bulk density (SOL_BD), available water capacity of the soil layer
(SOL_AWC), saturated hydraulic conductivity (SOL_K), organic carbon (SOL_CBN), clay
(CLAY), silt (SILT), sand (SAND), and rock fragment (ROCK) contents, moist soil albedo
(SOL_ALB), erodibility factor (USLE_K), and electrical conductivity (SOL_EC). Beyond the
columns describing layered soil information, there are 20 columns (i.e., columns 133 to 152)



describing two variables [i.e., soil CaCO$_3$ (SOL_CA) and soil pH (SOL_PH)] for 10 soil layers.
These variables are not currently active in SWAT and are assigned a value of zero.

## 4.  Merging the two datasets

Despite its usefulness as a source of soil information for hydrological simulations, the SLC
dataset is not assembled in a format readable by SWAT or other similar models. For example,
SWAT stores all the properties for a specific soil in a single row in the the '*usersoil*' table, while
this information is stored in the SLC as multiple rows in two different tables (i.e., SNT and
SLT). Thus, the information contained in the SLT database has to be processed to satisfy
SWAT's format requirements. In addition, all properties in the *usersoil* are spatially defined
while those of SLC are often stored in a multi-polygon structure with no unique spatial
identification. Variables required by SWAT and contained in the dataset presented here were
either extracted from SNT and SLT, or calculated from the information therein. Some other
variables were estimated from published values. Extraction or calculation of variables was done
through an R code that imported both SNT and SLT, screened the data for missing records and
missing SWAT-required information (data screening is described in section 5), and sequentially
populated unique soil records in the database. This section describes how these variables were
defined.

## 5.  Data screening

*5.1 Screening out incomplete soil information in the SNT*
The use of the SNT is necessary as it links the soils information to the GIS coverage
containing the PAT. However, a first screening was required to remove soils from the SNT that
are not present in the SLT, as soil layer information is required by SWAT. The mismatch among



soils in both tables can occur for a number of reasons. For example, there are soils in both tables
that pedologists have identified but their properties have not yet been characterized. Also, soils
listed in one table may be absent from another table due to changes in soil classification. Finally,
soils listed as unclassified in the SNT (i.e., variable KIND=U) do not have any data associated
with them in the SLT and do not occur on any published map.
Out of the 14,063 unique soils in the SNT, 489 soils were missing in the SLT and, therefore,
removed from the analysis. These 489 soils correspond to around 3.5 % of the soils listed in the
SNT. Most of the missing soils were reported as "unclassified" (305 soils; 62.2 %), suggesting
that these soils have been identified, but their properties have not yet been characterized. Mineral
soils corresponded to 29.4 % (144 soils) of the total, likely a reflection of changes in
classification. The other two classes comprised non-true soils (e.g. mine tailings, urban land; 33
soils; 6.7 %) and organic soils (8 soils; 1.6 %). Also, only 58 of the 489 missing soils (11.0 %)
could be mapped through linking with the CMP table, making it impossible to do any spatial
analysis on the distribution of these soils across the country. However, since the SNT assigns a
province for each soil, it is possible to identify where these missing records occur. Most of the
missing soils were in British Columbia (167 soils; 34.2 %), Manitoba (151 soils; 30.9 %), and
Saskatchewan (133 soils; 27.2 %), with smaller proportions in Yukon (13 soils; 2.7 %), Ontario
(11 soils; 2.3 %), Nova Scotia (9 soils; 1.8 %) and Newfoundland (5 soils; 1.0 %).
*5.2 SWAT requirements*
The SWAT data requirements were used as a second level of screening to build the present
dataset. The soil input variables in SWAT can be either required or optional (Table 2; Arnold et
al., 2013). Required variables that could not be calculated or estimated (e.g., SOL_BD, SOL_K,
SOL_CBN, CLAY, SILT, and SAND) were used to separate complete from incomplete records.



Soils in the SLT containing or allowing derivation of all the variables required by SWAT were
compiled in a dataset comprising 11,838 unique soils that were importable into the model. Soils
in the SLT with missing records (i.e., variables entered as -9 in the database) for the required
SWAT variables (gray rows in Table 2) were removed from the analysis. These soils were
compiled into a soils list provided as a reference.

As for the non-matching soils in the SNT and SLT, only 547 out of 1736 (i.e., 31.5 %) soils

with missing information could be mapped through linking with the CMP table, which renders
any spatial representation of these soils unmeaningful. However, the provinces where these soils
occur could also be identified. The highest proportions of soils with incomplete information were
in British Columbia (490 soils; 28.2 %), Manitoba (391 soils; 22.54 %). Ontario (182 soils;
10.5 %) and Alberta (180 soils; 10.4 %) had intermediate values, while Newfoundland (123
soils; 7.1 %), Saskatchewan (102 soils; 5.9 %), New Brunswick (93 soils; 5.4 %), the Northwest
Territories (80 soils; 4.6 %), Nova Scotia (47 soils; 2.7 %), Quebec (30 soils; 1.7 %), and Yukon
(17 soils; 1.0 %) had  less than 10 % of the soils missing information.
**6.  Populating the user soil table in SWAT**

The variables in SWAT's 'usersoil' table refer to record indexing and soil classification, as

well as soil properties pertaining to the entire profile or specific layers. The variables in each of
these groups are described in the following sub-sections. The 'usersoil' table starts with a
number of columns that define the database and soil classification variables, followed by soil
profile and layer information, and inactive soil properties (Table 2).



*6.1 Database and soil classification variables*
The SWAT soil classification variables include the OBJECTID (general listing number),
MUID (map unit identifier), SEQN (sequence number), SNAM (soil name), S5ID (Soils5-ID
number for USDA soil series data) and CMPPCT (percentage of the soil component in the
MUID). A numbering system used for the OBJECTID variable was chosen to avoid conflicts
with existing soils in the user soil table. The SWAT model comes with more than 200 soils in a
built-in database that cannot be easily overwritten, and any soils imported into the database with
the same OBJECTID as existing soils will not be imported. Thus, the OBJECTID field was
populated sequentially from 1001 to the number of unique soils in the SLC database plus 1000
(i.e., OBJECTID ends in 12,838 in the case of the COMPLETE dataset, which has 11,838 unique
soils). The map unit ID (MUID) was assigned the SOIL_ID code in the SLC dataset, which is a
concatenation of the province code (two digits), a soil code (three digits), a modifier code (five
digits), and a profile code (one digit). The sequence number (SEQN) variable was assigned the
same value as the OBJECTID variable. This process created a unique SEQN for each recurrence
in the SLC dataset.
Similar to the MUID variable, the soil name variable (SNAM) was also assigned the
SOIL_ID code in the SLC, despite the soil name being in the database, so as to link the soil
information to the GIS layer. The S5ID variable was created as a concatenation between the
acronym "SLC" and the province two-digit abbreviation code. For example, all the soils in the
province of Alberta have S5ID equal to "SLCAB". The CMPPCT variable was assigned a value
of 100, meaning that the soil comprises 100 % of this component. As stated in section 2, the user
has to make a decision on how to handle multipart polygons in the pre-processing of the SLC
GIS dataset since the soils in multi-component polygons are not spatially defined.



*6.2 Soil profile information*
The following six variables in the dataset (i.e., columns 7 to 12) pertain to soil profile
information. The number of layer variables (NLAYERS) was defined according to the soil layers
in the SLT below the soil surface. The SLT table also contains information for layers above the
soil surface as is the case of litter, which have negative values for upper and lower depths (i.e.,
the ground surface corresponded to the zero depth, while above surface and below surface layers
have negative and positive values, respectively). Above-surface layers were removed from the
dataset prior to analysis through filtering layers with lower depth above the soil surface (i.e.,
lower depth less than or equal to zero).
The hydrologic soil group (HSG) variable (HYDGRP) is an influential parameter for
estimation of runoff using the SCS-Curve Number method and, consequently, for hydrological
simulations in SWAT (Gao et al., 2012; Neitsch et al., 2005). The HSGs were calculated
according to the method outlined by USDA-NRCS (1993), which is based on depth to the
impermeable layer (e.g., bedrock), depth from soil surface to shallowest water table during the
year, hydraulic conductivity of the least conductive layer of the soil profile, and depth range of
the hydraulic conductivity. The specific criteria used are provided in tabular form as
supplementary material. Soils in the dual HSG classes were assigned to the less restrictive class
since most agricultural soils in Canada exhibit some degree of drainage (e.g., municipal drainage
network, surface drains, or tile drainage). SWAT translates HSG alphabetical classification into a
numeric system, where HSGs A, B, C, and D, are interpreted as 1, 2, 3, and 4, respectively. The
runoff potential increases with increasing numeric designations.
The depth to the impermeable layer is not reported in the SLC database and was estimated
based on the soil layers available in the SLT. When a bedrock layer or specific soil horizons



were present [i.e., fragipan; duripan; petrocalcic; orstein; petrogypsic; cemented horizon; densic
material; placic; bedrock, paralithic; bedrock, lithic; bedrock, densic; or permafrost; USDA-
NRCS (1993)], its upper depth was used as the depth to impermeable layer. When a bedrock
layer was absent, the lower depth of the deepest mineral soil layer was used as an alternative.
The shallowest annual depth to water table is also not reported and was estimated based on
drainage class reported in the SNT. Very poorly drained, poorly drained, imperfectly drained,
moderately well drained, and well drained (or better) soils were assigned water table depths of 0 ,
25 , 75, 100, and 125 cm, respectively. The variables pertaining to hydraulic conductivity of the
least conductive layer of the soil profile and depth range of the hydraulic conductivity were both
calculated using information from the SLT.
Out of the 11,838 soils in the generated dataset, 21.3, 24.6, 39.0, and 15.1 % belonged to
HSGs 1, 2, 3 and 4, respectively. These results suggest that more than half of the soils in Canada
have a relatively high or high runoff generation potential (i.e., HSGs 3 and 4, respectively). A
spatial analysis indicated that 20.0, 26.8, 36.7, and 16.5% of the areal extend of the soils
belonged to HSGs 1, 2, 3, and 4, respectively. Much of the soils with higher potential for runoff
generation are in the humid regions of  Ontario, Quebec, and the Maritimes (Fig. 2). Not
surprisingly, this region has extensively adopted measures to address excess moisture in
agricultural soils, such as tile drainage (Stonehouse, 1995; Rasouli et al., 2014). Excess moisture
is also a problem in areas of Canadian Prairies, such as the Red River Valley in Manitoba, where
surface drainage (Bower, 2007) and a growing use of  tile drainage  (Cordeiro and Sri Ranjan,
2012, 2015) have been used to address this problem. Conversely, soils with low potential for
runoff generation are located in Saskatchewan and Southeastern Alberta (along the
Saskatchewan border), which are among the most arid regions in Canada (Wolfe, 1997).



The maximum rooting depth of the soil profile (SOL_ZMX) was assumed to be the lower
depth of the deepest layer in the SLC soil profile. The fraction of soil porosity from which anions
are excluded (ANION_EXCL) was not available in the SLC database and was set to the default
value of 0.5 in SWAT (Arnold et al., 2013). This variable affects the concentration of nitrate in
the mobile water fraction, which is directly related to nitrate leaching. The potential of maximum
crack volume of the soil profile expressed as a fraction of the total soil volume (SOL_CRK) can
be calculated by the FLOCR model using 30-yr weather data (Bronswijk, 1989). However, due
to the fact that the model is not readily available for download and the unreasonable time
required to run the model for such a large number of soil types, as well as the fact that
SOL_CRK is optional in SWAT, its value was set of 0.5. In large scale studies this value is
further adjusted through a spatially explicit calibration scheme (Whittaker et al., 2010). The
SOL_CRK variable controls the potential crack volume for the soil profile. This value was
selected based on the fact that all of the built-in soils in the SWAT soils database have the
SOL_CRK variable set to 0.5. The TEXTURE variable, although not required for simulations
with the SWAT model, was estimated for reference using the 'TT.points.in.classes' function
from the 'soiltexture' R package (Moeys, 2016). The Canadian soil texture classification system
was used as a reference.
*6.3 Soil layer information*
The soil profile variables are followed by 10 sets of 12 variables (i.e., columns 13 to 132)
pertaining to layered soil information. The lower depth of each soil layer in the SLT was used as
the depth from soil surface to the bottom layer (SOL_Z). The soil bulk density (SOL_BD) was
extracted directly from the SLT. The available water capacity of the soil layer (SOL_AWC) was
calculated from the water retention of the soil reported in the SLT at different matric potentials.





The water moisture content at -33 and -1500 kPa were assumed to represent the soil moisture at
field capacity (FC) and permanent wilting point (PWP), respectively (Givi et al., 2004). The
SOL_AWC was calculated as the difference between FC and PWP (Hillel, 1998). Soil moisture
content at -33 kPa was not available for 2,658 layer records (i.e., 4.3% of the 61905 original
records in the SLT table), which would result in the variable SOL_AWC not being calculated
and the loss of more soils from the dataset. To avoid this, the moisture content at -10 kPa was
used to replace that at -33 kPa. On average, the soil moisture content in the soil profile was
around 6 mm larger at -10 kPa than that at -33 kPa (Table 3), indicating an overestimation of
SOL_AWC in these soils. Larger differences between soil moisture content at -10 kPa and -33
kPa in the top soil layers were likely driven by lower bulk densities, which increase the water
holding capacity of the soil (Table 3).

The variables saturated hydraulic conductivity (SOL_K) and soil organic carbon content

(SOL_CBN), as well as the clay (CLAY), silt (SILT), sand (SAND), and rock fragment (ROCK)
contents, were extracted directly from the SLT. The moist soil albedo (SOL_ALB) variable was
only required for the top layer as subsequent layers were assigned a value of zero. Since this
variable is not reported in the SLC database, it was estimated as the average (i.e., 0.10) of the
range reported by Maidment (1993) for moist, dark, plowed fields (i.e., 0.05-0.15). Again, this
value was selected since the SLC version 3.2 focuses on agricultural areas, which is also the
major domain simulated by SWAT.

Another important variable for SWAT is the erodibility factor (USLE_K), used as an input to

the Universal Soil Loss Equation (USLE). This equation is used to calculate soil erosion, which
is inherently linked to sediment and nutrient transport (Sharpley et al., 1992; He et al., 1995;
Sharpley et al., 2002; Aksoy and Kavvas, 2005; Koiter et al., 2013) and therefore, critical for



simulations of non-point sources of pollution. The erodibility factor was calculated using the
method presented by Sharpley and Williams (1990), which is based on the sand, silt, clay, and
organic carbon content of the soil (Eq. 1):

$$K = \left(0.2 + 0.3 \cdot \exp\left[-0.256 \cdot m_s \cdot \left(1 - \frac{m_{silt}}{100}\right)\right]\right) \cdot \left(\frac{m_{silt}}{m_c + m_{silt}}\right)^{0.3} \cdot$$

$$\left(1 - \frac{0.25 \cdot orgC}{orgC + \exp[3.72 - 2.95 \cdot orgC]}\right) \cdot \left(1 - \frac{0.7 \cdot \left(1 - \frac{m_s}{100}\right)}{\left(1 - \frac{m_s}{100}\right) + \exp\left[-5.51\_22.9 \cdot \left(1 - \frac{m_s}{100}\right)\right]}\right) \quad (1)$$

where $K$ is the erodibility factor [0.01 (ton·acre·hr)/(acre ft-ton in)], $m_s$ is the sand content
(percent), $m_{silt}$ is the silt content (percent), $m_c$ is the clay content (percent), and $orgC$ is the
organic carbon content (%) of the respective soil layer.

As for SOL_ALB, USLE_K is only required for the top layer and subsequent layers were

also assigned a value of zero. When converted from Imperial to SI units (Foster et al., 1981), the
range of calculated values (Table 4) generally agrees with the ranges reported for Canada (Wall
et al., 2002), taking into consideration that K values may vary, depending on particle size
distribution, organic matter, structure and permeability of individual soils (Wall et al., 2002).
However, the units in the dataset presented here were kept in Imperial units for consistency with
the SWAT input format. The spatial distribution of the erodibility factor (Fig. 3) was anticipated
to align with HSG, which was the case in areas of low erosion potential in Saskatchewan where
sandy soils prevail and in areas where runoff potential is high such as in southern Ontario.
However, the spatial distribution of USLE_K somewhat contrasted to that of HSG in some areas
of Manitoba and British Columbia, where low sediment transport potential was predicted in areas
with high runoff potential. This contrast was likely due to other factors reducing the potential for





sediment transport, such as soils with high clay to silt ratios or high organic carbon contents
(Sharpley and Williams, 1990).

The soil electrical conductivity (SOL_EC) information was extracted directly from the SLT.

The last twenty columns of the dataset (i.e., columns 133 to 152), which correspond to
SOL_CAL for the 10 soil layers followed by SOL_PH for the same layers, were all populated
with zeros since these variables are not currently active in SWAT. These variables also had
values of zero for all the pre-existing soils in the built-in database in the model.

**7.  Importing the SLC dataset into SWAT database**

Although the SWAT database is in a proprietary format (i.e., Microsoft Access), the present

soils dataset has been published in a non-proprietary format [i.e., comma-separated values (CSV)
file] that can be opened in a variety of software packages. However, the dataset can be easily
imported into the SWAT soils database using an automated import routine in Microsoft Access.
This import process consists of opening the SWAT2012 database and using the 'Import Text
File' tool under the 'Import & Link' section of the 'External Data' tab to read the CSV file. This
action will prompt a window where the user can select the path to where the present dataset is
stored and specify how and where the data is stored in the database. The option 'Append a copy
of the record to the table' should be selected, which activates a drop-down menu from which the
'usersoil' table should be highlighted. Once these options have been processed, an 'Import Text
Wizard' window will be prompted, where the option 'Delimited – Characters such as comma or
tab separate each field' should be selected. Processing of this selection will prompt another
window where the option 'comma' should be automatically selected by the wizard. However, the
user should activate the box 'First Row Contains Field Names' since the first row of the present



dataset contains the variable labels. Confirming the processing of the next windows should
finalize the import process, and the data should be ready to be used in SWAT predictions.
**8.  Data access**

PANGAEA, an open access library to archive, publish and distribute georeferenced data,

supports database-dependent research. Therefore, the entire dataset is published and archived in
the PANGAEA database (https://doi.pangaea.de/10.1594/PANGAEA.877298) under Creative
Commons Attribution 3.0 Unported, where the user must give appropriate credit, provide a link
to the license and indicate if changes are made.
**9.  Conclusions**

The soils dataset presented and discussed in this work represent an effort to facilitate

hydrological simulations using the SWAT model in Canada. The dataset consists of a
compilation of 11,838 different soils from the SLC database with all the information required by
SWAT and is ready to be imported into the model's soils database. A two-level data screening
procedure removed 489 soils with missing layered information (i.e., not present in the SLT),
while 1,736 soils were removed due to the lack of  critical information required by SWAT, such
as soil bulk density or saturated hydraulic conductivity. Among the major contributions of this
dataset, the calculation and/or estimation of variables not reported in the SLC database are of
special importance. The hydrologic soil groups (HSGs) calculated from SLC database suggests
that about half of the soils in Canada belong to classes with higher potential to generate runoff
(i.e., HSG classes 3 and 4). Occurrence of soils in HSG 3 and 4 agree with management practices
aimed at addressing excess moisture conditions in agricultural fields, such as subsurface drainage
in southern Ontario and Manitoba. The erodibility factor, which is another important variable for
SWAT simulations of non-point source pollution, suggest a relationship with runoff potential in



portions of  southern Ontario and Nova Scotia. However, low erodibility potential likely driven
by high clay to silt ratios or high organic carbon content were found in areas with higher runoff
potential in Manitoba and British Columbia.
**Author contribution**
M.R.C Cordeiro and R. Kroebel developed the concept for development of the dataset. G.
Lelyk interpreted the soil information contained in the SLC database. M.R.C Cordeiro and G.
Lelyk developed the methodology for deriving the soil variables. M.R.C Cordeiro developed the
code using R programming language to process the SLC dataset and performed data analysis. All
the authors revised the dataset and participated in manuscript preparation.
**Acknowledgements**
This research was supported by Beef Cattle Research Council and Agriculture and Agri-Food
Canada through the Beef Cluster, Environmental Footprint of Beef Project and the Alberta
Livestock and Meat Agency (ALMA) of the Alberta Agriculture and Forestry (Grant #
2016E017R).

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



Table 1. Description of variables in SWAT's 'usersoil' table.

| Variable Group | Column number in 'usersoil' table | Variables[a] |
|---|---|---|
| Database indexing | 1 | OBJECTID |
| Soil classification | 2 through to 6 | MUID; SEQN; SNAM; S5ID; CMPPCT |
| Soil properties | | |
| Profile | 7 trough to 12 | NLAYERS; HYDGRP; SOL_ZMX; ANION_EXCL; SOL_CRK; TEXTURE |
| Layers | 13 through to 132 (12 variables for 10 soil layers) | $SOL\_Z_x$; $SOL\_BD_x$; $SOL\_AWC_x$; $SOL\_K_x$; $SOL\_CBN_x$; $CLAY_x$; $SILT_x$; $SAND_x$; $ROCK_x$; $SOL\_ALB_x$; $USLE\_K_x$; $SOL\_EC_x$ |
| Inactive | 133 through to 152 | $SOL\_CAL_x$; $SOL\_PH_x$ |

[a] Subscript x corresponds to soil layer from 1 to 10.





Table 2. Variables included in the SWAT user soil table.

| Column | Variable[a] | Description | Units | Status |
|---|---|---|---|---|
| 1 | OBJECTID | Object identifier | – | Optional |
| 2 | MUID | Mapping unit identifier | – | Optional |
| 3 | SEQN | Record counter calculated by SWAT | – | Optional |
| 4 | SNAM | Soil identifying name | – | Optional |
| 5 | S5ID | Soil interpretation record | – | Optional |
| 6 | CMPPCT | Soil component percent | – | Optional |
| 7 | NLAYERS[†] | Number of layers | – | Required |
| 8 | HYDGRP | Hydrologic Soil Group | – | Required |
| 9 | SOL_ZMX | Maximum rooting depth of the soil profile | mm | Required |
| 10 | ANION_EXCL | Fraction of soil porosity from which anions are excluded | – | Optional |
| 11 | SOL_CRK | Potential of maximum crack volume of the soil profile expressed as a fraction of the total soil volume | $mm^3\ mm^{-3}$ | Optional |
| 12 | TEXTURE | Texture of soil layer | – | Optional |
| 13 | $SOL\_Z_x$ | Depth from soil surface to bottom of layer | mm | Required |
| 14 | $SOL\_BD_x$ | Moist bulk density | $Mg\ m^{-3}$ or $g\ cm^{-3}$ | Required |
| 15 | $SOL\_AWC_x$ | Available water capacity of the soil layer | $mm\ mm^{-3}$ | Required |
| 16 | $SOL\_K_x$ | Saturated hydraulic conductivity | $mm\ h^{-1}$ | Required |
| 17 | $SOL\_CBN_x$ | Organic carbon content | % (w/w) | Required |
| 18 | $CLAY_x$ | Clay content | % (w/w) | Required |
| 19 | $SILT_x$ | Silt content | % (w/w) | Required |
| 20 | $SAND_x$ | Sand content | % (w/w) | Required |
| 21 | $ROCK_x$ | Rock fragment content | % (w/w) | Required |
| 22 | $SOL\_ALB_x$ | Moist soil albedo | – | Required |
| 23 | $USLE\_K_x$ | Erodibility factor (K) | 0.01 (ton·acre·hr)/(acre ft-ton in) | Required |
| 24 | $SOL\_EC_x$ | Electrical conductivity | $dS\ m^{-1}$ | Optional |

Adapted from Arnold et al. (2013) and Sheshukov et al. (2009). [a] Subscript x corresponds to soil layer from 1 to 10. The variables $SOL\_CAL_x$ and $SOL\_PH_x$ are
present in the user soil table after all the columns listed above for all the 10 pre-existing layers. These variables refer to soil $CaCO_3$ and soil pH, respectively, and
are not currently active in the model. Thus, their records are entered zero in the SWAT 2012 database. [†]The number of layers defines how many entries will be
required in the layered information, signalled by the subscript x. For example, a soil with NLAYERS=4 should have subscript x corresponding to soil layer
variables from 1 to 4. As a result, the records extend to column 60 in the user soil table. (i.e., 4 layers×12 variables + 12 preceding variables=60).





Table 3. Average soil moisture content at matric potentials -10 kPa and -33kPa and average soil bulk density for
discrete layers of the soil profile. The average was calculated for all soils in the dataset. Each layer could have
different depths for individual soils used in the average.

| Layer | $\overline{\theta}\,at-10kPa$ | $\overline{\theta}\,at-33kPa$ | Difference (mm) | Average soil bulk density (g cm$^{-3}$) |
|---|---|---|---|---|
| 1 | 36.8 | 29.67 | 7.13 | 1.13 |
| 2 | 33.65 | 26.72 | 6.93 | 1.27 |
| 3 | 31.99 | 25.36 | 6.63 | 1.38 |
| 4 | 29.48 | 23.32 | 6.16 | 1.47 |
| 5 | 28.1 | 22.17 | 5.93 | 1.50 |
| 6 | 27.26 | 21.53 | 5.73 | 1.52 |
| 7 | 27.03 | 21.42 | 5.61 | 1.54 |
| 8 | 26.98 | 21.17 | 5.81 | 1.54 |
| 9 | 25.05 | 18.86 | 6.19 | 1.55 |
| AVERAGE | 29.59 | 23.36 | 6.24 | 1.43 |

$\overline{\theta}$ = average soil moisture content (mm).



Table 4. Comparison between the average erodibility factor (K) calculated for
each soil textural class in the SWAT dataset and values reported in the
literature.

| Soil Textural Class | Acronym | Calculated average K | Reported K range[†] |
|---|---|---|---|
| Loam | L | 0.14 | 0.23 – 0.30 |
| Heavy clay | HCl | 0.18 | 0.05 – 0.23 |
| Silty clay loam | SiClLo | 0.22 | 0.30 – 0.38 |
| Clay loam | ClLo | 0.14 | 0.23 – 0.30 |
| Silt loam | SiLo | 0.22 | 0.30 – 0.38 |
| Sand | Sa | 0.04 | < 0.05 |
| Sandy loam | SaLo | 0.11 | 0.05 – 0.23 |
| Clay | Cl | 0.14 | 0.23 – 0.30 |
| Silty clay | SiCl | 0.22 | 0.23 – 0.30 |
| Loamy sand | LoSa | 0.07 | < 0.05 |
| Sandy clay loam | SaClLo | 0.10 | 0.23 – 0.30 |
| Silt | Si | 0.55 | 0.30 – 0.38[¶] |
| Sandy clay | SaCl | 0.09 | 0.05 – 0.23[#] |

[†]Adapted from Wall et al. (2002). [¶]Range not reported; value from SiLo
used. [#] Range not reported; value from SaLo used.






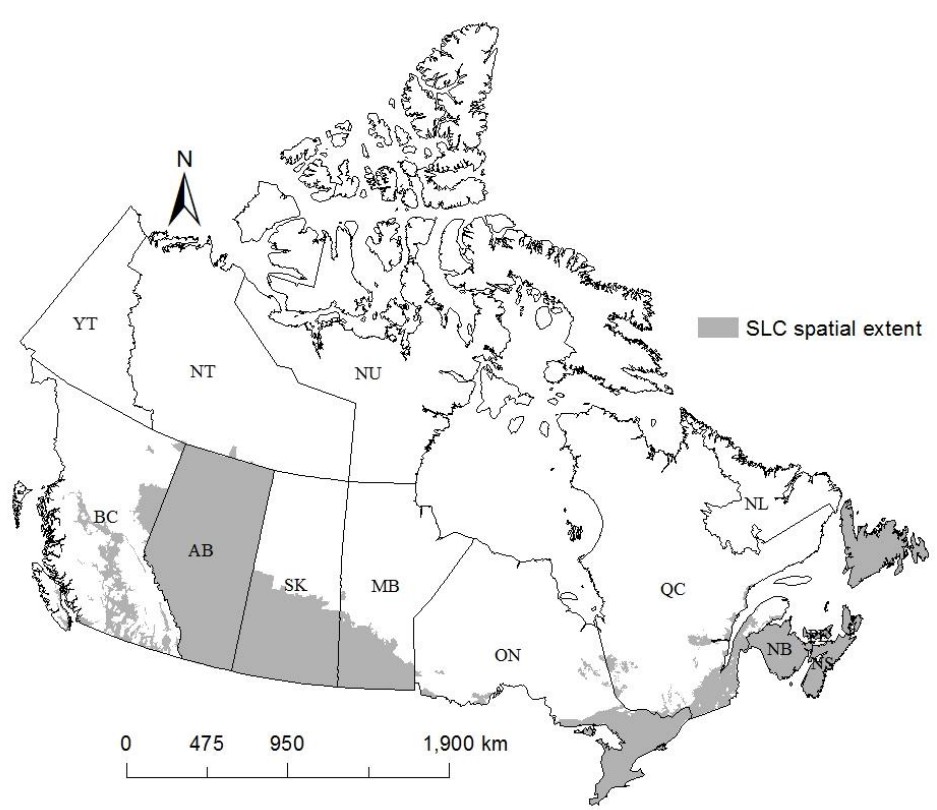

Figure 1. Spatial extent of the Soil Landscapes of Canada (SLC) database showing coverage in the Provinces of
Newfoundland and Labrador (NL), Prince Edward Island (PE), Nova Scotia (NS), New Brunswick (NB), Quebec
(QC), Ontario (ON), Manitoba (MB), Saskatchewan (SK), Alberta (AB), and British Columbia (BC), as well as the
Northwest Territories (NT).




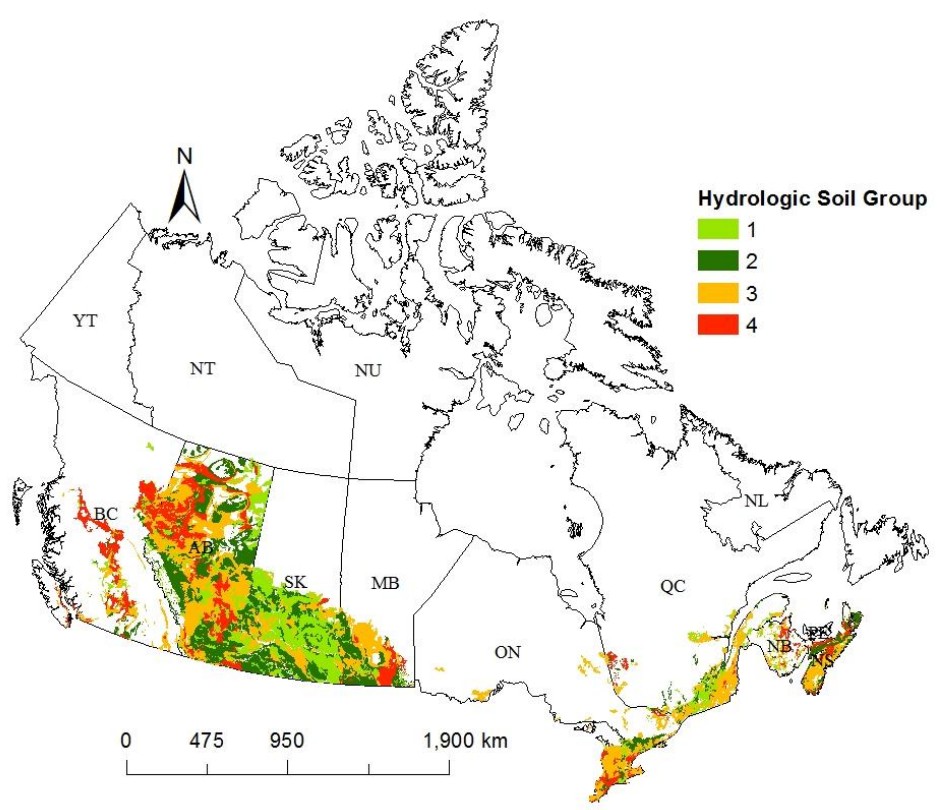

Figure 2. Spatial distribution of the hydrologic soil groups (HYDGRP) variable calculated for the Soil Landscapes
of Canada (SLC) database. HSG A=1, HSG B=2, HSG C=3, and HSG D=4 shown for the Provinces of Prince
Edward Island (PE), Nova Scotia (NS), New Brunswick (NB), Quebec (QC), Ontario (ON), Manitoba (MB),
Saskatchewan (SK), Alberta (AB), and British Columbia (BC). Some HSG could not be mapped [e.g. Province of
Newfoundland and Labrador (NL)] due to missing records in the PAT of the GIS layer or being part of the soils with
missing data in the SLT.

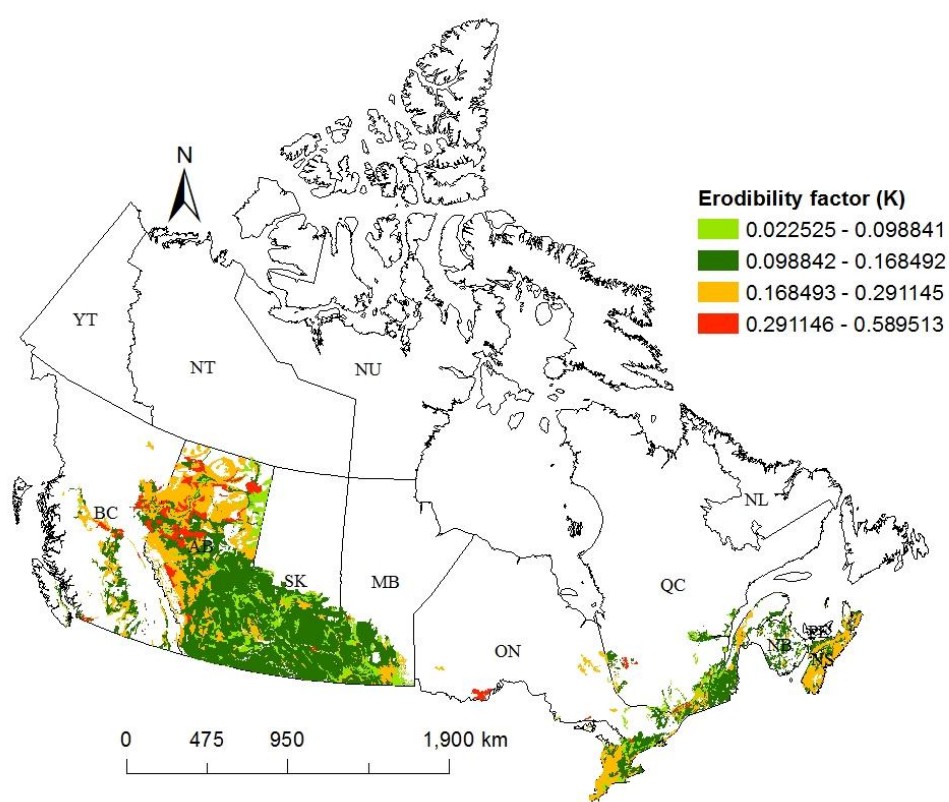

Figure 3. Spatial distribution of the erodibility factor (K) calculated for the Soil Landscapes of Canada (SLC)
database (Imperial units). The K factor shown for the Provinces of Prince Edward Island (PE), Nova Scotia (NS),
New Brunswick (NB), Quebec (QC), Ontario (ON), Manitoba (MB), Saskatchewan (SK), Alberta (AB), and British
Columbia (BC). Some HSG could not be mapped [e.g. Province of Newfoundland and Labrador (NL)] due to
missing records in the PAT of the GIS layer or being part of the soils with missing data in the SLT.