# Peer review of "Deriving a country-wide soils dataset from the Soil Landscapes of Canada (SLC) database"

_Earth System Science Data, 2017_

## Referee Comment (RC1) · Anonymous Referee #1 · 6 Dec 2017

Review ESSD-2017-66

Canadian soils data set

Summary: Perhaps and probably a notable and skilful effort by Canadian soil scientists. However, persistent and pervasive sloppiness about terminology substantially diminishes the quality of the work as described. The manuscript needs a section on uncertainties; some information exists in individual data processing sections but the overall description lacks a clear message to potential users about strengths and weaknesses of this approach and of the data as presented. This reviewer found, surprisingly, no indication of larger impact, e.g. to the community of SWAT users outside of Canada. Based on inherent deficiencies and absence of demonstrated impact this reviewer feels inclined to recommend that the editors reject this dataset and associated description for ESSD. However, considering the ESSD mission to promote open data access and the evident and positive motivation of these data providers to meet that goal, I would accede to an editorial decision to accept subject to very major revisions.

Data easy to access via Pangaea. Authors should please use the full doi citation, e.g. https://doi.pangaea.de/10.1594/PANGAEA.877298, to allow one-click access? But the downloads on the Pangaea landing page only lead to metadata. One needs to open the metadata files to find actual access, e.g. at http://store.pangaea.de/Publications/CordeiroM-etal_2017/SLCsoils_Canada_SWAT_Cordeiro-etal_2017.csv. The .csv files download successfully but one needs to go through 2 or 3 steps to access them. Could the authors work with Pangaea to reduce the number of steps required for access?

Already in the abstract we encounter this phrase: "14,063 unique soils". I don't believe the authors can document much less distinguish more than 14 thousand unique soils. I believe they might have 14k separate samples or profiles, but by any classification system (and one wonders why this paper did not follow the standard Canadian soil classification system, e.g. http://sis.agr.gc.ca/cansis/taxa/cssc3/index.html) these 14,000 individual profiles must fall into far fewer soil types?

Page 2, lines 44 to 46: Here the authors have grossly underestimated the use and application of SWAT. A Google Scholar search on SWAT model use and validation shows more than 7000 entries since 2013, covering agricultural soils in Africa, Asia, Europe and the Americas. If the authors hope to convince users that this effort has widespread value, e.g. for other agricultural systems across Ukraine, Hungary, China, or Australia, they need to first demonstrate that they understand the breadth of SWAT use.

Page 3, lines 51 to 53. Finally, mention of global impact of SWAT but tied to one old (2010) reference? No evidence from this group to show they understand widespread use nor anticipate widespread interest in their work.

Page 3, lines 59 to 62. Data only for USA? Researchers in no other country have undertaken a similar effort to the one described here? This reviewer doubts very much that the work presented here represents the only non-USA effort.

Page 3, lines 63 to 70. Here the authors list roughly 15 SWAT uses by the ag community of Canada alone. So SWAT does have wider impact among Canadian researchers, but what about its wider global impact? We don't need an inventory of international SWAT applications but we do need recognition and evidence that this work has impact beyond Canada's borders.

Page 4, lines 82 to 85 and Figure 1. Even if these areas represent all or even the majority of Canadian ag soils, this delineation falls far short of the phase in the title "country-wide". A more accurate description would say many of the agriculturally-relevant soils along the southern fringe of the country. Whatever the final wording, the title must change.

Page 4, line 88: "all of Canada's agricultural soils". We need evidence of the reliability of this statement. Figure 1 casts substantial doubt.

Page 5, line 98: "a country-level soils data set".  This phrase seems more accurate than earlier phrases.

Page 5, lines 102, 103: "Such a laborious pre-processing exercise is widely, but inconsistently adopted in SWAT simulations reported in the literature."  And?  So?  Does the effort described here have relevance to prior and future SWAT simulations outside of Canada?  One hopes so, but the authors fail to make that point either as a preface or as an outcome.

Terminology confusion: "unique soils", "soil types", "soil series", "individual soil and landscape features", "soil component", "soil", "soil variables", soil properties", "unique soil records", "soils", "soils have been identified, but their properties have not yet been characterized", "soil profile and layer information", "inactive soil properties", "soil profile information", "different soils".  The authors throw these various terms around casually, but without consistent intention? 'Soil types' used  perhaps most frequently among all these terms and phrases but never defined.  Even less definition for the other terms.  Again, why have we apparently wandered so far from Canada's own soil classification system and why do we find little or no consistency with global (WoSIS) efforts described below?  If, in the end, these authors focus on the hydrological properties (e.g. HSG: hydrological soil group), they should establish a clearer or more consistent terminology for the less essential (in this context) details?  The manuscript could start from the goal of determining hydologic erodability and then proceed through a series a assembly and compilation steps starting with and adhering to a careful definition of terms?

This inconsistent terminology develops into acute confusion on page 11 (section 6.1).  Here the SWAT model contains 200 predefined "soils".  In order to avoid non-recognition of additional imported data, the Canadian researchers have manipulated their ID codes to allow ingest of more than 11k additional "unique soils".  But we get zero information to confirm that the soil categorisation (level of detail of description) of the extant SWAT soil data matches the level of detail of the added 11k "unique" soils.  One hopes and suspects that these researchers have matched apples with apples here, but they give us no evidence to show that they have not matched SWAT apples with SLC oranges.   The 11k number invites concern.  I used STATSGO2 for Montana, a state that borders and in fact shares watersheds with a substantial portion of the SLC regions.  If I  assume that the field MUKEY equates to what one might consider in terms of this activity "soil type", I find more than 4300 entries (soil samples?, soil profiles?) but only approximately 700 distinct MUKEY values.  Just over the boarder to the north, we find 11,000 'entries' or 11,000 distinct soil 'types'?

The World Soil Information Service (WoSIS) records more than 90000 **profiles** ( https://doi.org/ 10.5194/essd-9-1-2017) consisting (with the inclusion of depth / layer information) of more than 4 million records.  They manage this international data deluge by adhering to analytical and physical properties described in the *GlobalSoilMap* (http://www.globalsoilmap.net) specifications.  A reader wishes that this Canadian effort had adopted the same approach!  Or, if they have followed the same approach despite their confusing terminology, they need to inform users and readers!

Page 13, line 284: "more than half of the soils in Canada".  Not even close to true!  Half of the soils under agricultural use for which profile data exist in a small southern portion of Canada!

Page 17, Section 7.  The information conveyed here works very poorly or even not at all as a narrative.  Make this into a flow chart or a bulleted list of steps?

Page 18, line 398: "Among the major contributions of this dataset, the calculation and/or estimation of variables not reported in the SLC database are of special importance." Perhaps true and perhaps this represents a substantial outcome of the skill and effort of this group.  But does it have any relevance beyond this relatively small subset of Canadian soils?  If the authors contend that it does have larger relevance, they have not made the case!  If it does not have larger broader global relevance, does it belong in ESSD?

---

## Author Comment (AC1) · 26 Feb 2018

Authors' reply to interactive comment posted by Anonymous Referee #1 regarding the ESSD Discussion paper "Deriving a country-wide soils dataset from the Soil Landscapes of Canada (SLC) database for use in Soil and Water Assessment Tool (SWAT) Simulations" (essd-2017-66).

Dear Referee,

We appreciate your comments and suggestions to strengthen the manuscript. Please find below the answers to your comments.

*1. Reviewer: Summary: Perhaps and probably a notable and skilful effort by Canadian soil scientists. However, persistent and pervasive sloppiness about terminology substantially diminishes the quality of the work as described. The manuscript needs a section on uncertainties; some information exists in individual data processing sections but the overall description lacks a clear message to potential users about strengths and weaknesses of this approach and of the data as presented. This reviewer found, surprisingly, no indication of larger impact, e.g. to the community of SWAT users outside of Canada. Based on inherent deficiencies and absence of demonstrated impact this reviewer feels inclined to recommend that the editors reject this dataset and associated description for ESSD. However, considering the ESSD mission to promote open data access and the evident and positive motivation of these data providers to meet that goal, I would accede to an editorial decision to accept subject to very major revisions.*

Authors: This was in fact a large effort to bring the SLC dataset into the SWAT format. The terminology has been changed for consistency in the revised manuscript (please see answers to comments below). A section on uncertainty has also been included to discuss strengths and weaknesses of the dataset. Regarding impact, the reviewer is correct that the purpose of this dataset is directed to SWAT applications in Canada and its direct impact is limited to SWAT users in other regions, except for providing a description of the steps that can be duplicated to support similar efforts elsewhere. However, a quick search on the ESSD database reveals a number of papers describing local or regional datasets[1,2,3,4,5,6], which also have reduced impact to the broad scientific community outside those areas. Moreover, the dataset is a long-due effort to support SWAT simulations in Canada. This need has been recognized and discussed with fellow researchers doing SWAT modelling in this country and prompted the effort described in the

[1] Garel, E., and Ferreira, Ó.: Multi-year high-frequency physical and environmental observations at the Guadiana Estuary, Earth System Science Data, 7, 299-309, 2015.
[2] Fayad, A., Gascoin, S., Faour, G., Fanise, P., Drapeau, L., Somma, J., Fadel, A., Al Bitar, A., and Escadafal, R.: Snow observations in Mount Lebanon (2011–2016), ibid., 9, 573, 2017.
[3] Petersen, G. N.: Meteorological buoy measurements in the Iceland Sea, 2007–2009, ibid., 779.
[4] Serrano-Notivoli, R., Beguería, S., Saz, M. Á., Longares, L. A., and de Luis, M.: SPREAD: a high-resolution daily gridded precipitation dataset for Spain–an extreme events frequency and intensity overview, ibid., 721.
[5] Chen, G., Pan, S., Hayes, D. J., and Tian, H.: Spatial and temporal patterns of plantation forests in the United States since the 1930s: an annual and gridded data set for regional Earth system modeling, ibid., 545.
[6] Wood, C. M., Smart, S. M., Bunce, R. G., Norton, L. R., Maskell, L. C., Howard, D. C., Scott, W. A., and Henrys, P. A.: Long-term vegetation monitoring in Great Britain-the Countryside Survey 1978-2007 and beyond, ibid., 445-459.

manuscript. The authors believe that the concerns rose by the reviewer (i.e. terminology and uncertainty) are fully addressable and that the dataset merits publication by ESSD, despite its limited geographical scope. The value of this endeavour is further supported in that we have already been approached by SWAT users regarding utilizing the developed dataset. This would suggest that the dataset will make a clear contribution to the scientific community.

*2. Reviewer: Data easy to access via Pangaea. Authors should please use the full doi citation, e.g. https://doi.pangaea.de/10.1594/PANGAEA.877298, to allow one-click access? But the downloads on the Pangaea landing page only lead to metadata. One needs to open the metadata files to find actual access, e.g. at http://store.pangaea.de/Publications/CordeiroM-etal_2017/SLCsoils_Canada_SWAT_Cordeiro-etal_2017.csv. The .csv files download successfully but one needs to go through 2 or 3 steps to access them. Could the authors work with Pangaea to reduce the number of steps required for access?*

Authors: The DOI citation has been updated in the revised manuscript. The download process has been modified with PANGEA and the dataset can also be downloaded through a single click.

*3. Reviewer: Already in the abstract we encounter this phrase: "14,063 unique soils". I don't believe the authors can document much less distinguish more than 14 thousand unique soils. I believe they might have 14k separate samples or profiles, but by any classification system (and one wonders why this paper did not follow the standard Canadian soil classification system, e.g. http:// sis.agr.gc.ca/cansis/taxa/cssc3/index.html) these 14,000 individual profiles must fall into far fewer soil types?*

Authors: The SWAT model treats each entry in the database as a "soil" and this terminology was followed here. The "14,063 unique soils", therefore, refer to unique records in the SLC database. These records are a concatenation of the Province acronym, soil code, modifier, and profile type. However, to comply with the reviewer's request, the terminology has been revised throughout the manuscript and now reads "unique soil records".

*4. Reviewer: Page 2, lines 44 to 46: Here the authors have grossly underestimated the use and application of SWAT. A Google Scholar search on SWAT model use and validation shows more than 7000 entries since 2013, covering agricultural soils in Africa, Asia, Europe and the Americas. If the authors hope to convince users that this effort has widespread value, e.g. for other agricultural systems across Ukraine, Hungary, China, or Australia, they need to first demonstrate that they understand the breadth of SWAT use.*

Authors: The authors are aware of the vast SWAT literature. However, these sentences mentioned by the reviewer were not intended to convince the users of the widespread value of the dataset discussed here. As stated in the reply to comment #1, the dataset is meant to be used in Canada. To highlight importance of the SWAT model for hydrologic simulations in Canada, a number of regional studies were cited in the manuscript. To emphasize the wide adoption of the

model elsewhere, a single reference[7] with more than 240 citations has been used to summarize the importance of SWAT in the original submission, as opposed to a more explicit description of applications in different continents. By using this approach, the authors were trying to be cognizant of the manuscript length, which was more than 8,000-words long in the original submission. However, to comply with the reviewer's suggestion, two newer references[8,9] have been cited in the revised manuscript. The Arnold et al. (2012) reference has 645 citations, while the Gassman et al. (2014) reference has 123 citations, which succinctly highlight the importance of the SWAT model.

*5. Reviewer: Page 3, lines 51 to 53. Finally, mention of global impact of SWAT but tied to one old (2010) reference? No evidence from this group to show they understand widespread use nor anticipate widespread interest in their work.*

Authors: As stated in the answer to comment # 4, a single reference[7] with more than 240 citations has been used in the original submission to summarize the importance of SWAT, as opposed to a more explicit description of applications in different continents. The single reference classified as "old" by the reviewer is a landmark paper bringing an overview of the SWAT model and deals with important aspects such as development, model parameterization and uncertainty, as well as practical applications including calibration and validation. This reference was a deliberate choice for readers and potentials SWAT users to assess essential information in a single source. Although the authors are aware of the widespread use of SWAT, they are trying to emphasize the importance of the dataset to the audience in Canada. However, to comply with the reviewer's suggestion, two newer references[8,9] have been cited in the revised manuscript. The Arnold et al. (2012) reference has 645 citations, while the Gassman et al. (2014) reference has 123 citations, which succinctly highlight the widespread adoption of the SWAT model.

*6. Reviewer: Page 3, lines 59 to 62. Data only for USA? Researchers in no other country have undertaken a similar effort to the one described here? This reviewer doubts very much that the work presented here represents the only non-USA effort.*

Authors: SWAT comes with a built-in soils database for the USA only. Other efforts outside the USA have to generate/compile/pre-process soil information from several sources to some extent, which is time consuming. Most SWAT modelling exercises outside the USA had to go through this process, but most likely, only for the area of interest. The purpose of this dataset is to alleviate this requirement regarding soils in most agricultural regions of Canada. The authors are
* * *
[7] Douglas-Mankin, K. R., Srinivasan, R., and Arnold, J. G.: Soil and Water Assessment Tool (SWAT) Model: Current Developments and Applications, 53, 10.13031/2013.34915, 2010.

[8] Arnold, J. G., Moriasi, D. N., Gassman, P. W., Abbaspour, K. C., White, M. J., Srinivasan, R., Santhi, C., Harmel, R. D., Griensven, A. v., Liew, M. W. V., Kannan, N., and Jha, M. K.: SWAT: Model Use, Calibration, and Validation, Transactions of the ASABE, 55, 1491-1508, 2012.

[9] Gassman, P. W., Sadeghi, A. M., and Srinivasan, R.: Applications of the SWAT model special section: Overview and insights, Journal of Environmental Quality, 43, 1-8, 10.2134/jeq2013.11.0466, 2014.

unaware of any similar effort trying to pre-process an entire country-wide soils dataset to offer a product that is ready to be used in SWAT simulations. The fact that we have already had enquires regarding use of the dataset, demonstrates that the Canadian scientific community will make use of the dataset.

*7. Reviewer: Page 3, lines 63 to 70. Here the authors list roughly 15 SWAT uses by the ag community of Canada alone. So SWAT does have wider impact among Canadian researchers, but what about its wider global impact? We don't need an inventory of international SWAT applications but we do need recognition and evidence that this work has impact beyond Canada's borders.*

Authors: As stated in answers to comments # 1, #4, and #5, the dataset is meant to be used in Canada. Due to that, emphasis was placed on the literature on SWAT applications in Canada. The authors believe that the number of references (roughly 15, as pointed out by the reviewer), is appropriate for that description and to keep the length of the manuscript in balance. However, to comply with the reviewer's request, the paragraph has been extended and now cites the two new references[8,9] mentioned in previous answers. The revised text now reads:

> "Worldwide, SWAT has emerged as one of the most widely used water quality watershed- and river basin-scale models for simulation of a broad range of hydrologic and/or environmental problems (Gassman et al., 2014). These applications in different regions are described in the extensive body of peer-reviewed SWAT literature (Arnold et al., 2012). Specifically in Canada, the SWAT model has been used for hydrological simulations in most provinces, including…"

*8. Reviewer: Page 4, lines 82 to 85 and Figure 1. Even if these areas represent all or even the majority of Canadian ag soils, this delineation falls far short of the phase in the title "country-wide". A more accurate description would say many of the agriculturally-relevant soils along the southern fringe of the country. Whatever the final wording, the title must change.*

Authors: The SLC dataset represents the agricultural regions of Canada. The wording and the title of the revised manuscript have been modified to highlight this aspect, as suggested.

*9. Reviewer: Page 4, line 88: "all of Canada's agricultural soils". We need evidence of the reliability of this statement. Figure 1 casts substantial doubt.*

Authors: The wording has been changed to "most soils across the agricultural regions of Canada" in the revised manuscript.

*10. Reviewer: Page 5, line 98: "a country-level soils data set". This phrase seems more accurate than earlier phrases.*

Authors: This terminology has been kept in the revised manuscript.

*11. Reviewer:* *Page 5, lines 102, 103: "Such a laborious pre-processing exercise is widely, but inconsistently adopted in SWAT simulations reported in the literature." And? So? Does the effort described here have relevance to prior and future SWAT simulations outside of Canada? One hopes so, but the authors fail to make that point either as a preface or as an outcome.*

Authors: This sentence was included in the manuscript to highlight the uniqueness and novelty of the dataset. However, it is not meant to suggest that a global soils dataset has been derived for SWAT. This sentence has been removed in the revised manuscript to avoid any ambiguous interpretation. Our work does illustrate a framework that could be used to develop datasets that would be appropriate for regionally or nationally specific modelling efforts in other areas of the world.

*12. Reviewer:* *Terminology confusion: "unique soils", "soil types", "soil series", "individual soil and landscape features", "soil component", "soil", "soil variables", soil properties", "unique soil records", "soils", "soils have been identified, but their properties have not yet been characterized", "soil profile and layer information", "inactive soil properties", "soil profile information", "different soils". The authors throw these various terms around casually, but without consistent intention? 'Soil types' used perhaps most frequently among all these terms and phrases but never defined. Even less definition for the other terms. Again, why have we apparently wandered so far from Canada's own soil classification system and why do we find little or no consistency with global (WoSIS) efforts described below? If, in the end, these authors focus on the hydrological properties (e.g. HSG: hydrological soil group), they should establish a clearer or more consistent terminology for the less essential (in this context) details? The manuscript could start from the goal of determining hydologic erodability and then proceed through a series a assembly and compilation steps starting with and adhering to a careful definition of terms?*

Authors: As stated in the answer to comment #3, the terminology has been revised throughout the manuscript and now reads "unique soil records". Noteworthy, the present dataset had to adhere to the SWAT convention for naming the variables in order to make it fully compatible and importable into the model's soil database. As the SWAT terminology differs from that of WoSIS, it follows that the terminology in the manuscript cannot be consistent with that used by WoSIS. Translating the SLC soils terminology into SWAT's terms was a major part of the pre-processing analysis [the SLC database is built on Canadian National Soil Database (NSDB) standards as maintained by Canadian Soil Information Service (CanSIS)]. Also, the objective of the dataset was to provide a complete dataset since every parameter is important for SWAT simulations. Hydrologic Soils Groups and Erodibility factor (K) are only discussed in more detail because they were not present in the original SLC dataset and had to be calculated. Thus, they represent a novel contribution worth being discussed.

*13. Reviewer:* *This inconsistent terminology develops into acute confusion on page 11 (section 6.1). Here the SWAT model contains 200 predefined "soils". In order to avoid non-recognition*

*of additional imported data, the Canadian researchers have manipulated their ID codes to allow ingest of more than 11k additional "unique soils". But we get zero information to confirm that the soil categorisation (level of detail of description) of the extant SWAT soil data matches the level of detail of the added 11k "unique" soils. One hopes and suspects that these researchers have matched apples with apples here, but they give us no evidence to show that they have not matched SWAT apples with SLC oranges. The 11k number invites concern. I used STATSGO2 for Montana, a state that borders and in fact shares watersheds with a substantial portion of the SLC regions. If I assume that the field MUKEY equates to what one might consider in terms of this activity "soil type", I find more than 4300 entries (soil samples?, soil profiles?) but only approximately 700 distinct MUKEY values. Just over the boarder to the north, we find 11,000 'entries' or 11,000 distinct soil 'types'?*

Authors: Understanding this section of the manuscript requires some knowledge about database structure for both SLC and SWAT. The OBJECTID variable is the one used to index each entry in the SWAT database (this variable is not present in the SLC database). Since SWAT has 200 built-in soils records and potentially more if the user has already previously imported any, the OBJECTID in the present dataset started at 1001. This numbering creates a margin of safety around 800 entries to avoid conflict with any pre-existing soil record. There is no level of detail required by SWAT, as long as the OBJECTID is unique and does not conflict with the existing soils. The importability of the dataset was tested in that regard. There should be no concerns regarding a match between SWAT and SLC since the OBJECTID variable is only applicable to SWAT and not to SLC. The authors would like to clarify that not all variables present in the SWAT database are present in the SLC database and vice-versa. Thus, the dataset at hand does not preserve all variables in the SLC but only those required by SWAT. It also makes use of information present in the SLC to calculate variables required by SWAT. Regarding the second part of the comment, the MUKEY or MUID variables refer to map units (i.e. polygon features in GIS datasets). A single map unit can contain from one to many soil components within it (1:1 or 1:N) relationships in the soils database. That is, the soil within a map unit may not be homogeneous and the components represent different entries in the databased with contrasting physical-chemical properties. That is why there are 4,300 soil components but only 700 distinct MUKEYs in the example the reviewer mentioned (i.e. on average, around six soil components in each map unit). For SWAT applications using the present dataset, which is the focus here, the MUID is irrelevant since it is not used in the model calculations or for indexing. It was only kept here because it is a variable in the SWAT database, although not influential for simulations (the MUID variable may be used for importing soil information if the USA databases are used; that is, only for applications within the USA). For mapping purposes in SWAT, the field SNAM in the present dataset will likely be chosen by the user for indexing. For the example in Montana, the STATSGO2 database terminology is not consistent with that of SLC. The MUKEY variable in the former database is equivalent to POLY_ID variable in the SLC. Although these two variables refer to the map unit, they are standardized. This is an example of how the terminology

changes among datasets and that it is not always consistent with specific conventions (e.g. WoSIS).

*14. Reviewer: The World Soil Information Service (WoSIS) records more than 90000 profiles ( https://doi.org/10.5194/essd-9-1-2017) consisting (with the inclusion of depth / layer information) of more than 4 million records. They manage this international data deluge by adhering to analytical and physical properties described in the GlobalSoilMap (http://www.globalsoilmap.net) specifications. A reader wishes that this Canadian effort had adopted the same approach! Or, if they have followed the same approach despite their confusing terminology, they need to inform users and readers!*

Authors: The CanSIS (Canadian Soil Information System) that developed the SLC database does maintain a database adhering to the GlobalSoilMap specifications and is also a cooperating institution with WoSIS. Unfortunately, SWAT does not follow the same conventions. If so, the effort described here would not be necessary. The "confusion" referred by the reviewer actually results from translating the data from the well-established conventions into the database terminology defined by the SWAT programmers. Please note that the goal here is to present a dataset compatible with SWAT's terminology. Therefore, it is inevitable that the terminology used in the manuscript will depart from accepted standards from renowned classification systems.

*15. Reviewer: Page 13, line 284: "more than half of the soils in Canada". Not even close to true! Half of the soils under agricultural use for which profile data exist in a small southern portion of Canada!*

Authors: The sentence refers only to the soils included in the dataset, which covers the agricultural domain in Canada. Geographically, this domain is for the most part restricted to the southern portion of the country. The sentence has been reworded to avoid misinterpretations and now reads "more than half of the agricultural soil records in Canada".

*16. Reviewer: Page 17, Section 7. The information conveyed here works very poorly or even not at all as a narrative. Make this into a flow chart or a bulleted list of steps?*

Authors: A flow chart has been included in the revised manuscript, as suggested.

*17. Reviewer: Page 18, line 398: "Among the major contributions of this dataset, the calculation and/or estimation of variables not reported in the SLC database are of special importance." Perhaps true and perhaps this represents a substantial outcome of the skill and effort of this group. But does it have any relevance beyond this relatively small subset of Canadian soils? If the authors contend that it does have larger relevance, they have not made the case! If it does not have larger broader global relevance, does it belong in ESSD?*

Authors: As stated in answers to comments # 1, #4, #5, and #7, the dataset is meant to be used in Canada. The authors strongly believe that the dataset fits the ESSD scope as the journal does not focus solely on global datasets and has published papers describing local or regional datasets[1,2,3,4,5,6]. We have all had indications of a Canadian demand for this dataset in the form of enquires and directions to its use.

---

## Referee Comment (RC2) · Anonymous Referee #2 · 3 Jun 2018

This manuscript is proposing a great solution to many SWAT users around the World by following the Canadian example where a national soil database is available of from the FAO Harmonized World Soil Database elsewhere. The creation of the Canadian Soil SWAT database is very well described in much useful details for future users of this dataset, but also for those who would like to replicate the process in another country. This is a very valuable work and publication that will allow SWAT users in Canada to save valuable time and improve the calibration of their model.

As a SWAT user, not a soil scientist, I find therefore this work very useful and valuable

for publication of course for future Canadian users, but also for those willing to do similar work in other countries. I wish that more scientists around the world would make available such effort to save everybody a lot of time in constructing SWAT Soil database.

My main specific comment were joining those from the previous referee on the discussion of usefulness beyond Canadian users. I would therefore still recommend to better review and report on similar efforts elsewhere.

Otherwise, I find the paper ready for publication.

---

## Author Comment (AC2) · 4 Jun 2018

Authors' reply to interactive comment posted by Anonymous Referee #2 regarding the ESSD Discussion paper "Deriving a country-wide soils dataset from the Soil Landscapes of Canada (SLC) database for use in Soil and Water Assessment Tool (SWAT) Simulations" (essd-2017-66).

General Comments

Reviewer: This manuscript is proposing a great solution to many SWAT users around

the World by following the Canadian example where a national soil database is available of from the FAO Harmonized World Soil Database elsewhere. The creation of the Canadian Soil SWAT database is very well described in much useful details for future users of this dataset, but also for those who would like to replicate the process in another country. This is a very valuable work and publication that will allow SWAT users in Canada to save valuable time and improve the calibration of their model. As a SWAT user, not a soil scientist, I find therefore this work very useful and valuable for publication of course for future Canadian users, but also for those willing to do similar work in other countries. I wish that more scientists around the world would make available such effort to save everybody a lot of time in constructing SWAT Soil database.

Authors: The authors thank the reviewer for the comments. This effort was actually motivated by past experience with SWAT in Canada and by the lack of such dataset in that country. It is our hope that the dataset presented in this manuscript will save resources and promote hydrological simulations using SWAT in Canada. The authors also expect that the description provided in the manuscript will support similar undertakings elsewhere.

Specific Comments:

Reviewer: My main specific comment were joining those from the previous referee on the discussion of usefulness beyond Canadian users. I would therefore still recommend to better review and report on similar efforts elsewhere. Otherwise, I find the paper ready for publication.

Authors: Efforts similar to this one have been discussed in the revised manuscript.

---

## Short Comment (SC1) · 30 Jun 2018

The manuscript explains a great deal of effort that is spent to prepare SWAT-ready soil database in Canada. SWAT is a well-developed hydrological model capable of simulating management activities in agricultural lands and therefore is a very useful tool for analyzing land-use change scenarios. In fact, personally, I prepared a similar dataset for Southern Ontario; therefore, I know the value of the dataset in the larger scale, Canada-wide. Hence, I support the study and the dataset. I have two main comments that can help improve the usefulness of the dataset and the coherence of

the manuscript:

1. The study lacks a comparison between the proposed soil dataset for Canada and the SWAT database for US across the border. Such a comparison will make users aware of the discrepancy between the two datasets and give them a source of reference when modelling bi-national watersheds. Moreover, authors need to discuss the reason for the discrepancy between the two datasets across the border.

2. Authors need to discuss the uncertainty in the SWAT model structure for bi-national watersheds that use US soil dataset for the US side and the proposed soil dataset for the Canadian side of the watershed.

---

## Author Comment (AC3) · 19 Jul 2018

Authors' reply to interactive comment posted by Dr. Masoud Asadzadeh regarding the ESSD Discussion paper "Deriving a country-wide soils dataset from the Soil Landscapes of Canada (SLC) database for use in Soil and Water Assessment Tool (SWAT) Simulations" (essd-2017-66).

Dear Dr. Asadzadeh, We appreciate your comments and suggestions to strengthen the manuscript. Please find below the answers to your comments.

[Figure]

General comments Dr. Asadzadeh: The manuscript explains a great deal of effort that is spent to prepare SWAT-ready soil database in Canada. SWAT is a well-developed hydrological model capable of simulating management activities in agricultural lands and therefore is a very useful tool for analyzing land-use change scenarios. In fact, personally, I prepared a similar dataset for Southern Ontario; therefore, I know the value of the dataset in the larger scale, Canada-wide. Hence, I support the study and the dataset.

Authors: SWAT is a largely used model, as you state. However, the lack of ready-to-use datasets in Canada hinders its application in this country and requires great efforts as the one you describe to use the model in Southern Ontario. The major reason behind the present dataset was to provide users with one of the major inputs for SWAT model (i.e. soils data) and disseminate its adoption in Canada. We have already received compliments from users expressing their appreciation for our efforts in this regard.

Specific comments

I have two main comments that can help improve the usefulness of the dataset and the coherence of the manuscript:

1. Dr. Asadzadeh: The study lacks a comparison between the proposed soil dataset for Canada and the SWAT database for US across the border. Such a comparison will make users aware of the discrepancy between the two datasets and give them a source of reference when modelling bi-national watersheds. Moreover, authors need to discuss the reason for the discrepancy between the two datasets across the border.

Authors: Discrepancies among the national datasets had been identified even before the present analysis was carried out. These discrepancies are due to several reasons. The main one is that the American and Canadian datasets are populated with soils data consistent with their respective national methods and standards. As a result of data and methods differences in many areas across the international border, discontinuity between the datasets will almost always be shown. Other reasons for

discrepancies between the national datasets pertain to GIS mapping. Discontinuity between the GIS layers of both datasets (using a polygon topology) is possible, which would result in abrupt changes in soil properties at the border between the two countries. Also, soil records not spatially defined within multi-component polygons in the GIS layer compound the issue depending on which component is mapped (multi-component polygons are discussed in the manuscript). Even if methods standardization and GIS discontinuity were not an issue, disagreement between the datasets at the interface between the datasets could be due to differences in accuracy and precision in the measurements leading to parameter uncertainty (a new section in the revised manuscript discusses uncertainty in the present dataset). All these aspects, coupled with the large number of physico-chemical parameters in the datasets and the extent of the boundary between the two countries (i.e., more than 6,000 km), renders the comparison between the databases a very complex and time consuming analysis, which is out of the scope of the present study.

2. Dr. Asadzadeh: Authors need to discuss the uncertainty in the SWAT model structure for bi-national watersheds that use US soil dataset for the US side and the proposed soil dataset for the Canadian side of the watershed.

Authors: We believe that SWAT's model structure uncertainty is not dependent on soil database (i.e., input data) but on the description (i.e., algorithms) used to simulate physical processes in the model. In other words, the model uses the same algorithm in both US and Canadian portions of trans-boundary watersheds. However, model response in trans-boundary watersheds might be different due to input data uncertainty caused by differences in soil data quality and quantity in US and Canadian datasets. The discussion suggested by the reviewer has been added to the revised manuscript in a newly added section dealing with uncertainty.